# Learning Abstract World Models for Value-preserving Planning with Options

## Abstract

General-purpose agents require fine-grained controls and rich sensory inputs to perform a wide range of tasks. However, this complexity often leads to intractable decision-making. Traditionally, agents are provided with task-specific action and observation spaces to mitigate this challenge, but this reduces autonomy. Instead, agents must be capable of building state-action spaces at the correct abstraction level from their sensorimotor experiences. We leverage the structure of a given set of temporally-extended actions to learn abstract Markov decision processes (MDPs) that operate at a higher level of temporal and state granularity. We characterize state abstractions necessary to ensure that planning with these skills, by simulating trajectories in the abstract MDP, results in policies with bounded value loss in the original MDP. We evaluate our approach in goal-based navigation environments that require continuous abstract states to plan successfully and show that abstract model learning improves the sample efficiency of planning and learning.

## 1 Introduction

Reinforcement learning (RL) is a promising framework for embodied intelligence because of its flexibility, generality, and online nature. Recently, RL agents have learned to control complex control systems: stratospheric balloons (Bellemare et al., 2020), nuclear fusion reactors (Degrave et al., 2022) and drones (Kaufmann et al., 2023). They have also mastered long-horizon decision-making problems such as the game of Go and chess (Silver et al., 2016; 2018). To achieve these results, each agent's state representation and action spaces were engineered to make learning tractable: the state space was designed to contain only relevant information for decision-making and the actions were restricted to task-relevant decisions to be made at every time step. This is in conflict with the state-action space required for versatile, general-purpose agents (e.g., robots), which must possess broad sensory data and precise control capabilities to handle a wide variety of tasks, such as playing chess, folding clothes or navigating a maze. Abstractions alleviate this tension: action abstractions enable agents to plan at larger temporal scales and state abstractions reduce the complexity of learning and planning; a combination of action and state abstraction results in a new task model that can capture the natural complexity of the task, instead of the complexity of the agent (Konidaris, 2019).

For instance, in model-based RL (MBRL; Sutton (1991); Deisenroth and Rasmussen (2011)), there is a long line of research that focuses on learning transition and reward models to plan by simulating trajectories. Many modern methods learn abstract state spaces (Ha and Schmidhuber, 2018; Zhang et al., 2019; Silver et al., 2018; Hafner et al., 2019; 2021; 2023) to handle complex observation spaces. However, they learn models for the primitive action spaces and work within the single-task setting. Recently, there has been interest in using MBRL for skill discovery: Hafner et al. (2022) learn a model in an abstract state space and learn a further abstraction over it to discover goals in a Feudal RL manner (Dayan and Hinton, 1992). Bagaria and Konidaris (2020); Bagaria et al. (2021a;b), instead, assume that the abstract state space is a graph and learn skills that connect nodes in that graph, effectively building a model that is both abstract in state and in actions. These approaches are ultimately limited because they assume a discrete abstract state space.

On the other hand, in robotics, high-level planning searches for sequences of *temporally-extended* actions (motor skills) to achieve a task. However, the agents needs a model to compute plans composed of their motor skills and this is typically given to the agent. To enable the agent to learn a model compatible with its motor skills from sensor data, Konidaris et al. (2018) propose novel

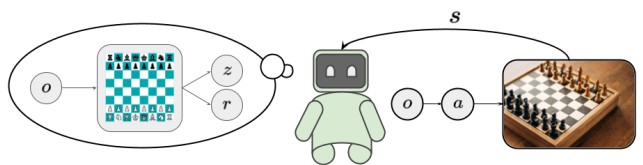

Figure 1: An agent needs to solve a task using its actuators and sensors (on the right). However, it requires an abstract model of the task (on the left) to reason at long time scales. This can be constructed by combining temporally-extended actions with a compatible abstract state representation that contains the minimal information necessary for planning with those actions.

semantics to automatically learn logical predicates from the agent observation space that support task planning with PDDL (Planning Domain Definition Language; Fox and Long (2003); Younes and Littman (2004)). Moreover, they provide theoretical guarantees for learning predicates that support sound task planning. In a similar vein, Ugur and Piater (2015a;b); Ahmetoglu et al. (2022) propose to cluster the effects of motor skills to build discrete symbols for planning. Similarly, Asai et al. (2022) introduce a discrete VAE (Variational Auto-encoder; (Kingma and Welling, 2013)) approach to leverage modern deep networks for grounding PDDL predicates and action operators from complex observations. While these approaches consider temporally-extended actions and are promising for planning problems where the appropriate state abstractions are discrete, they are not applicable when planning with the available high-level actions requires a continuous state representation.

Instead, we are interested in learning state abstractions that are continuous, compatible with modern deep learning methods, and that guarantee value-preserving planning with a set of given skills. Specifically, we focus on building abstract world models in the form of Markov decision processes (MDPs) that have abstract state and action spaces and, in contrast to previous approaches, provide a principled approach to characterize the abstract state space that ensures that planning in simulation produces a policy with expected value equal to that by planning with the original MDP. In summary, we (1) introduce the necessary and sufficient conditions for constructing an abstract Markov decision process sufficient for value-preserving planning for a given set of skills; (2) introduce an information maximization approach compatible with contemporary deep learning techniques, ensuring a bounded value loss when planning using the abstract model; and finally, (3) provide empirical evidence that these abstract models support effective planning with off-the-shelf deep RL algorithms in goal-based tasks (Mujoco Ant mazes (Fu et al., 2020) and Pinball (Konidaris and Barto, 2009) from pixels).

## 2 BACKGROUND AND NOTATION

**Markov Decision Processes** A continuous state, continuous action Markov decision process (MDP) (Puterman, 2014) is defined as the tuple $M = (\mathcal{S}, \mathcal{A}, T, R, p_0, \gamma)$ where $\mathcal{S} \subseteq \mathbb{R}^{d_s}$ is the state space and $\mathcal{A} \subseteq \mathbb{R}^{d_a}$ is the action space ($d_s, d_a \in \mathbb{N}$), $T : \mathcal{S} \times \mathcal{A} \to \Delta(\mathcal{S})$ is the transition kernel that represent the dynamics of the environment, $R : \mathcal{S} \times \mathcal{A} \to \mathbb{R}$ is the reward function bounded by $R_{Max} \in \mathbb{R}$, $\gamma \in [0, 1)$ is the discount factor and $p_0 \in \Delta(\mathcal{S})$ is the initial state distribution.

**Planning and Bellman Equation** A solution to an MDP is a policy $\pi : \mathcal{S} \to \Delta(\mathcal{A})$ that maximizes the expected return $J(\pi) = \mathbb{E}\left[\sum_{t=0}^{\infty} \gamma^t R(S_t, A_t) | S_0 \sim p_0, \pi, T\right]$. An important family of solution methods for MDPs are based on the Bellman optimality principle and the Bellman equation. For a given policy $\pi$, the state-value function $v^\pi : \mathcal{S} \to \mathbb{R}$ is defined as $v^\pi(s) := \mathbb{E}\left[\sum_{t=0}^{\infty} \gamma^t R(S_t, A_t) | S_0 = s, \pi\right]$. The state-value function represents the expected discounted return when following the policy $\pi$ from state $s$. Importantly, the value function satisfy the following recursion, known as the Bellman equation, which is used in many current planning and learning methods for MDPs: $v^\pi(s) = \mathbb{E}\left[R(s, a) + \gamma \int_{\mathcal{S}} T(s'|s, a) v^\pi(s') ds'\right]$.

**Action Abstractions** Options (Sutton et al., 1999) are a formalization of temporally-extended actions, or *skills*, that are used by the agent to plan with a longer temporal scope than that allowed by primitive actions. An option $o$ is defined by the tuple $(I_o, \pi_o, \beta_o)$ where $I_o : \mathcal{S} \to \{0, 1\}$ is the initiation set, that is, the set of states in which the option can start execution; $\pi_o$ is the policy function, and $\beta_o : \mathcal{S} \to [0, 1]$ is the termination probability function that indicates the probability of terminating the option execution at state $s$.

**Expected-length Model of Options** Generally, options are used to plan in Semi-Markov decision processes (SMDP; Sutton et al. (1999)), in which modelling jointly the option's dynamics $T$ and duration $\tau$ as $T_\gamma(s'|s,o) = \sum_{\tau=0}^{\infty} \gamma^\tau \Pr(S_\tau = s', \beta(s_\tau)|S_0 = s, o)$ and its reward as $R(s,o) = \mathbb{E}_\tau \left[\sum_{t=0}^{\tau-1} \gamma^t R(S_t, A_t)|s, o\right]$, result in the Multi-time model of options. However, we will use a simpler and more practical model of options dynamics: the expected-length model of options (Abel et al., 2019). In this case, the option's duration is modeled independently from the next-state distribution. More precisely, let $\tilde{\tau}_o$ be the average number of timesteps taken to execute the option $o$, then $T_\gamma(s'|s,o) = \gamma^{\tilde{\tau}_o} p(s'|s,o)$ where $p(s'|s,o)$ is the probability density function over the next-state observed when the option is executed as a black-box skill.

**State Abstractions and Probabilistic Groundings** State abstractions (or state aggregation) have commonly been defined in the form of surjective functions $f : \mathcal{S} \to \bar{\mathcal{S}}$ where $\bar{\mathcal{S}}$ is an abstract state space. Recently, Konidaris et al. (2018) propose probabilistic groundings to define a new class of state abstractions. These groundings are defined by $G : \bar{\mathcal{S}} \to \Delta(\mathcal{S})$ and, contrary to state aggregation approaches, these can have overlapping support. That is, for a state $s$ and abstract states $\bar{s}^1$ and $\bar{s}^2$, we can have that $G_{\bar{s}^1}(s) > 0$ and $G_{\bar{s}^2}(s) > 0$. In state aggregation methods, one state has just one abstract state to map to. Therefore, this provides a more expressive framework to build abstractions.

## 3 Value-preserving Abstract MDPs

To plan with a set of options, we must build a model of their effects. In this section, we formalize this model as an MDP with the following characteristics: (1) **Action Abstraction**, the action space is the set of task-relevant temporally-extended skills (i.e., the ground actions are not used for planning); (2) **State Abstraction**, because the set of skills operate at a higher-level of abstraction, the observation space will contains more information than required to plan with the skills; (3) **Sufficient for Planning**, the model must support computing a plan with the option set for task-specific rewards. In the case of abstract MDPs, the abstract model must guarantee *accurate* trajectory simulations to leverage the planning and RL algorithms developed for MDPs.

### 3.1 Ground and Abstract MDPs

We start by defining the ground MDP $M$, the environment that the agent observes by only executing the options.

**Definition 3.1** (Ground MDP). Let $\mathcal{O}$ be a set of options defined over the agent's state-action space. The ground MDP is $M = (\mathcal{S}, \mathcal{O}, T, R, \gamma, \tau, p_0)$. $T(s'|s,o)$ is the next-state probability density function seen by the agent when executing option $o$ at $s$ and its accumulated discounted reward is $R(s,o) = \mathbb{E}_\tau \left[\sum_{t=1}^{\tau} \gamma^t R(S_t, A_t)|s, o\right]$, and $\tau : \mathcal{S} \times \mathcal{O} \to [0, \infty)$ is the expected option's execution time of option $o$ when initiated at state $s$[1].

**Definition 3.2** (Abstract MDP). The abstract MDP is $\bar{M} = (\bar{\mathcal{S}}, \mathcal{O}, \bar{T}, \bar{R}, \gamma, \bar{\tau}, \bar{p}_0)$ where $\bar{\mathcal{S}}$ is the abstract state space, $\bar{T} : \bar{\mathcal{S}} \times O \to \Delta(\bar{S})$ is the abstract transition kernel, $\bar{R} : \bar{\mathcal{S}} \times \mathcal{O} \to \mathbb{R}$ is the abstract reward function, $\gamma$ is the discount factor, $\bar{\tau} : \bar{\mathcal{S}} \times \mathcal{O} \to [0, \infty)$ is the option's duration model and $\bar{p}_0$ is the initial abstract state distribution.

Given that the objective is to compute plans in the abstract model, we will only consider policies of the form $\pi : \bar{\mathcal{S}} \to \mathcal{O}$ in the rest of the paper. Moreover, to connect the abstract MDP to the ground MDP, we use a grounding function defined in terms of probability density functions, as introduced by Konidaris et al. (2018). The grounding of an abstract state $\bar{s}$ is defined by the probability of the agent being in a state $s$.

**Definition 3.3** (Grounding function). Let $M$ be a ground MDP and $\bar{M}$ be an abstract MDP. A grounding function $G : \bar{\mathcal{S}} \to \Delta(\mathcal{S})$ maps $\bar{s}$ to probability measures over $\mathcal{S}$ of $M$. Given an abstract state $\bar{s}$, we denote by $G_{\bar{s}}$ its grounding probability density. We will denote the tuple $(M, \bar{M}, G)$ as a grounded abstract model.

---

[1] The ground MDP would be an SMDP if we used the multi-time model of options (Sutton et al., 1999).

### 3.2 THE DYNAMICS PRESERVING ABSTRACTION

Our goal is to build an abstract model that enables the agent to simulate trajectories as though it had access to a simulator of the ground model. To achieve this, we establish two key distributions: the future state distribution and the grounded future state distribution.

**Definition 3.4** (Future State Distribution). Let the tuple $(M, \bar{M}, G)$ be a grounded abstract model. Let the future state distribution be $B_t$, and defined recursively as follows,

$$B_0(s_0) = p_0(s_0)$$
$$B_t(s_t, ..., s_0|o_0, ..., o_{t-1}) = T(s_t|s_{t-1}, o_{t-1})B_{t-1}(s_{t-1}, ..., s_0|o_0, ..., o_{t-2}).$$

and the grounded future state distribution $\bar{B}_t$ is the estimate obtained by grounding the estimate obtained by simulating trajectories in the abstract model $\bar{M}$

$$P(s_t, \bar{s}_t, ..., s_0, \bar{s}_0|o_0, ..., o_{t-1}) = G_{\bar{s}_t}(s_t)\bar{T}(\bar{s}_t|\bar{s}_{t-1}, o_{t-1})P_{t-1}(s_{t-1}, \bar{s}_{t-1}, ..., s_0, \bar{s}_0|o_0, ..., o_{t-2})$$

$$\bar{B}_t(s_t, ..., s_0|o_0, ..., o_{t-1}) = \int P(s_t, \bar{s}_t, ..., s_0, \bar{s}_0|o_0, ..., o_{t-1})d\bar{s}_0...\bar{s}_t.$$

Hence, we say that when $B_t(s_t|o_0, ..., o_{t-1}) = \bar{B}_t(s_t|o_0, ..., o_{t-1})$, then simulating a trajectory in the abstract model is the same as in the ground model. To satisfy this, we can build an abstract model based on dynamics-preserving abstractions[2].

**Definition 3.5** (Dynamics Preserving Abstraction). Let $\phi$ be a mapping $\phi : \mathcal{S} \to \mathcal{Z} \subseteq \mathbb{R}^{d_z}$ for some dimension $d_z \in \mathbb{N}$. If for all $o \in \mathcal{O}$ and all $s \in \mathcal{S}$ that are reachable, the following holds,

$$T(s'|s, o) = T(s'|\phi(s), o); \tag{1}$$
$$\Pr(I_o = 1|s) = \Pr(I_o = 1|\phi(s)); \tag{2}$$

where, $I_o$ is an indicator variable corresponding to the option's initiation set. Then, we say that $\phi$ is dynamics-preserving. That is, the information in $\phi(s)$ is sufficient to predict the effect of the option and to determine if an option is executable.

This is similar to model-preserving abstractions (Li et al., 2006) and bisimulation (Givan et al., 2003; Ferns et al., 2004). However, 1) it is stronger in the sense that $z$ must be a sufficient statistic for next-state prediction, and more importantly 2) this does not impose a condition over the ground reward function. Because we want to build an abstract model to be re-used for task-specific rewards (as we will see in Section 4.3), the ground reward function is considered as a way to measure the cost (negative reward) of executing a skill—retaining Markovianity with respect to the ground reward function would limit how much information can potentially be abstracted away.

We will now build a sensible abstract MDP $\bar{M}$, as follows. Let $\phi : \mathcal{S} \to \mathcal{Z}$ be a dynamics-preserving abstraction. Given that $T(s'|s, o) = T(s'|z, o)$ where $\phi(s) = z$, then we can build a transition function in $\mathcal{Z}$-space, $T(z'|z, o)$, and a grounding function $G$, that can let us reconstruct $T(s'|z, o)$.

$$p_0(z) = \int p_0(s)\mathbb{1}[\phi(s) = z]ds;$$

$$T(z'|z, o) = \int T(s'|z, o)\mathbb{1}[\phi(s') = z']ds';$$

$$G(s|z, o, z') = \begin{cases} \frac{p_0(s)\mathbb{1}[\phi(s)=z']}{p_0(z')} & \text{if } z' \text{ is an initial state (there is not previous } (z, o)) \\ \frac{T(s|z,o)\mathbb{1}[\phi(s)=z']}{T(z'|z,o)} & \text{otherwise} \end{cases};$$

Given that just knowing $z$ is not enough to determine its grounding distribution, we can build an abstract state space $\bar{\mathcal{S}} \triangleq \mathcal{Z} \times \mathcal{O} \times \mathcal{Z}$ of transition tuples—with special values values $z_\perp$ and $o_\perp$ to form $\bar{s}_0 = (z_\perp, o_\perp, z_0)$ for initial abstract states. Let $\bar{s} = (\hat{z}, \hat{o}, z)$ and $\bar{s}' = (\tilde{z}, \tilde{o}, z')$ be two abstract states in $\bar{\mathcal{S}}$, we define the abstract MDP functions in this new $\bar{\mathcal{S}}$, as follows.

$$G_{\bar{s}}(s) = G(s|z, o, z');$$

$$\bar{T}(\bar{s}'|\bar{s}, o) = \begin{cases} T(z'|z, o) & \text{if } \tilde{z} = z \text{ and } \tilde{o} = o \\ 0 & \text{otherwise} \end{cases};$$

$$\bar{R}(\bar{s}, o) = \mathbb{E}_{s \sim G_{\bar{s}}}[R(s, o)]; \quad \bar{\tau}(\bar{s}, o) = \mathbb{E}_{s \sim G_{\bar{s}}}[\tau(s, o)];$$

Finally, the following theorem formally states that this construction is sound.

---
[2]We defer all proofs to Appendix A.1.

**Theorem 3.6.** *Let the tuple $(M, \bar{M}, G)$ be a grounded abstract model and a function $\phi : \mathcal{S} \to \mathcal{Z} \subseteq \mathbb{R}^{d_z}$. The model satisfies that $B_t(\cdot \mid o_0, ..., o_{t-1}) = \bar{B}_t(\cdot \mid o_0, ..., o_{t-1})$ if and only if $\phi$ is dynamics-preserving.*

This theorem proves that if we learn a dynamics-preserving abstraction, we can simulate accurate trajectories in the abstract model. Therefore, planning in the abstract model is accurate, in the sense, that the value of an abstract state $v^\pi(\bar{s})$ computed using the abstract model is the same as the one would get by generating trajectories in the ground MDP and computing the expected value under grounding G, $\mathbb{E}_{s \sim G_{\bar{s}}}[v^\pi(s)]]$.

**Corollary 3.7.** *Let the tuple $(M, \bar{M}, G)$ be a grounded abstract model. If the dynamics preserving property holds then the value of policy $\pi$ computed in abstract model $\bar{M}$ satisfies that $v^\pi(\bar{s}) = \mathbb{E}[v^\pi(s)|s \sim G_{\bar{s}}]$. That is, the grounded abstract model preserves the expected value under the grounding G.*

*Proof.* Given that we have that, by definition, $T(s'|s, o) = T(s'|\bar{s}, o) = \mathbb{E}_{\bar{s}' \sim \bar{T}(\cdot|\bar{s}, o)}[G_{\bar{s}'}(s)]$. It follows that

$$
\begin{aligned}
\mathbb{E}_{s \sim G_{\bar{s}}}[v^\pi(s)] &= \mathbb{E}_{s \sim G_{\bar{s}}} \left[ \mathbb{E}_{o \sim \pi} \left[ R(s, o) + \mathbb{E}_{s' \sim T(s'|s, o)} \left[ \gamma^\tau v^\pi(s') \right] \right] \right] \\
&= \mathbb{E}_{o \sim \pi} \left[ \mathbb{E}_{s \sim G_{\bar{s}}} \left[ R(s, o) \right] + \mathbb{E}_{s \sim G_{\bar{s}}, s' \sim T(s'|s, o)} \left[ \gamma^\tau v^\pi(s') \right] \right] \\
&= \mathbb{E}_{o \sim \pi} \left[ \bar{R}(\bar{s}, o) + \mathbb{E}_{\bar{s}' \sim \bar{T}(\cdot|\bar{s}, o)} \mathbb{E}_{s' \sim G_{\bar{s}}} \left[ \bar{\gamma} v^\pi(s') \right] \right] \\
&= \mathbb{E}_{o \sim \pi} \left[ \bar{R}(\bar{s}, o) + \mathbb{E}_{\bar{s}' \sim \bar{T}(\cdot|\bar{s}, o)} \left[ \bar{\gamma} v^\pi(\bar{s}') \right] \right] = v^\pi(\bar{s}).
\end{aligned}
$$

$\square$

The Skills to Symbols framework (Konidaris et al., 2018) introduces the strong subgoal property to build grounded discrete symbols for sound classical planning. The next corollary proves that the strong subgoal is a special case of the dynamics preserving property when the appropriate abstraction function has finite co-domain. Therefore, we can build discrete dynamics preserving models if and only if the strong subgoal property holds.

**Corollary 3.8.** *Let the tuple $(M, \bar{M}, G)$ be a grounded abstract model. Let the strong subgoal property (Konidaris et al., 2018) for an option $o$ be defined as, $Pr(s'|s, o) = Pr(s'|o)$. The dynamics preserving property holds with a finite abstract state space $\mathcal{Z} = [N]$ for some $N \in \mathbb{N}$ if and only if the strong subgoal property holds.*

## 4 LEARNING THE ABSTRACT MODEL

### 4.1 INFORMATION MAXIMIZATION TO LEARN A DYNAMICS-PRESERVING $\phi$

The mutual information between random variables $X$ and $Y$, $MI(X; Y)$, measures the information that each variable holds about the other. We are interested in finding a function $\phi$ that is dynamics-preserving such that we can build our abstract MDP. By Definition 3.5, we want to learn $\phi(s)$ that is maximally predictive of the effect of $o$ when executed in $s$ and to predict if option $o$ is executable. That is, we want to maximize the following:

$$
\max_{\phi \in \Phi} MI(S', I; \phi(S), O) \equiv \max_{\phi \in \Phi} MI(S'; \phi(S), O) + MI(I; \phi(S)), \tag{3}
$$

where $\Phi$ is a class of functions that map the high-dimensional ground states to lower-dimensional space. $I$ is binary random variable for the initiation set prediction. $S', S, O$ are random variables over the ground states $\mathcal{S}$ and the options set $\mathcal{O}$.

In general, by the data processing inequality, $MI(S'; \phi(S), O)$ is upper-bounded by $MI(S'; S, O)$. Therefore, we can show that optimizing the above objective results in a bounded value loss when using the abstract model to plan. To see this, we first note that by compressing through $\phi$, we lose information $\Delta MI \triangleq MI(S'; S, O) - MI(S'; Z, O)$, where $Z = \phi(S)$, in the transition dynamics simulation. We show that,

$$
\Delta MI \stackrel{(a)}{=} \mathbb{E}_{p(s)} \left[ D_{KL} \left( T(s'|s, o) || \tilde{T}(s'|z, o) \right) \right] \stackrel{(b)}{\geq} 2 \ln 2 \mathbb{E}_{p(s)} \left[ \| T(s'|s, o) - \tilde{T}(s'|z, o) \|_1^2 \right].
$$

where $p(s)$ is a distribution over $s$ that will depend on the data collection policy and (a) follows from the definition of the KL divergence and (b) from the well-known bound relating the KL divergence and L1 norm[3]. Therefore, the error in the learned transition dynamics is minimized by our objective and this implies, by the following theorem, that this objective also minimizes the value loss resulting from the approximation.

**Theorem 4.1** (Value Loss Bound). *Let $(M, \bar{M}, G)$ be a grounded abstract model and $\tilde{T}(s'|\bar{s}, o) = \int G_{\bar{s}'}(s')\bar{T}(\bar{s}'|\bar{s}, o)d\bar{s}'$ be the approximate transition dynamics from the grounded model. If the following conditions hold for all $o \in \mathcal{O}$ and all $s \in \mathcal{S}$ with $G_{\bar{s}}(s) > 0$: (1) $\|T(s'|s, o) - \tilde{T}(s'|\bar{s}, o)\|_1^2 \leq \epsilon_T$, and (2) $|R(s, o) - \bar{R}(\bar{s}, o)|^2 \leq \epsilon_R$; then, for any policy $\pi$,*

$$|Q^\pi(s, o) - Q^\pi(\bar{s}, o)| \leq \frac{\sqrt{\epsilon_R} + \gamma V_{Max}\sqrt{\epsilon_T}}{1 - \gamma}.$$

### 4.2 CONTRASTIVE ABSTRACT MODEL LEARNING

We maximize the previous Infomax objective (3) as follows. The term $MI(I; Z)$ reduces to a cross entropy loss, so we will focus on estimating the term $MI(S'; Z, O)$: we can prove that maximizing both sides of the identity $MI(Z'; Z, O) = (MI(S'; Z') - MI(S'; Z'|Z, O))$ implicitly maximizes $MI(S'; Z, O)$ (see extended derivation details in Appendix A.2). Intuitively, the first term $MI(Z'; Z, O)$ makes $z'$ predictable from knowing the option executed and the previous $z$. The second term avoids collapsing $\phi$ to a trivial solution: maximizing $MI(S'; Z') - MI(S'; Z'|Z, O)$ makes $\phi$ retain information about the ground state $s$ (avoiding collapse of the representation) that is maximally predicted by the previous $(z, o)$.

We choose to maximize these mutual information terms contrastively using InfoNCE (Oord et al., 2018) to avoid making assumptions about tractable density models (other MI estimators (Poole et al., 2019; Alemi et al.; Belghazi et al., 2018) can be used). Using these estimators allows the model to implicitly learn complex grounding functions that improve the quality of the abstract state space. Note that by using InfoNCE for the terms above, this algorithm corresponds to Temporal Predictive Coding (TPC; Nguyen et al. (2021)) which proposes abstract states without reconstruction objectives. Therefore, our formulation approaches to the TPC algorithm in the degenerate case of options being the primitive actions[4].

---

**Algorithm 1** Planning and Learning with an Abstract Model

**Require:** Agent $\pi$, Ground Environment M, Abstract Model $\bar{M}$, Goal $\mathcal{G}$
1: Initialize dataset $\mathcal{D}$ by rolling out $N$ trajectories
2: $\bar{M} \leftarrow$ PretrainAbstractMDP($\mathcal{D}$)
3: $\bar{M} \leftarrow$ MakeTaskMDP($\bar{M}, \mathcal{G}$)
4: **while** true **do**
5:     $\mathcal{D} \leftarrow$ Roll out for $L$ steps.
6:     **if** $H$ steps have passed **then**
7:         $\bar{M} \leftarrow$ TrainModel($\bar{M}, \mathcal{D}$)
8:         $\pi \leftarrow$ TrainAgentImagination($\bar{M}, \pi$)
9:     **end if**
10: **end while**

---

In practice, we assume that we have access to a dataset of transition samples $\mathcal{D} = \{(s_i, o_i, r_i^\gamma, s_i', \tau_i, I_i)\}_{i=1}^N$ that correspond to the execution of option $o_i$ from state $s_i$, terminating in $s_i'$ with a duration of $\tau_i$ and accumulated return $r_i^\gamma = \sum_{t=0}^{\tau_i-1} \gamma^t r_t$. $I_i$ corresponds to the initiation sets of all options in state $s_i$. This dataset might be initialized by rolling out trajectories with a random agent and further enhanced during the agent's learning (see Algorithm 1).

We propose to learn the abstract model $M_\theta^\phi = (T_\theta^\phi, R_\theta, I_\theta^\phi, \tau_\theta)$ based on the abstraction $\phi$ parameterized by a function approximator $f_\phi$. Notice, that because we need to guarantee good initiation sets by $MI(I; \phi(S))$, the initiation set loss also affects the learning of $f_\phi$:

$$\mathcal{L}_\phi = -MI_\phi(Z'; Z, O) - MI_\phi(S'; Z');$$
$$\mathcal{L}_{\theta,\phi}^I = -w_i \log I_\theta^\phi(I_i|f_\phi(s_i));$$
$$\mathcal{L}_{\theta,\phi}^T = -\log \bar{T}_\theta(f_\phi(s_i')|f_\phi(s_i), o_i);$$

---

[3]$D_{KL}(P, Q) \geq 2\ln 2\|P - Q\|_1^2$
[4]Extended discussion in Appendix A.2

Therefore, $\mathcal{L}_\phi$, $\mathcal{L}^I_{\theta,\phi}$ and $\mathcal{L}^T_{\theta,\phi}$ are used to learn the abstraction function $f_\phi$. Moreover, we use a weighted negative log-likelihood loss for the initiation loss to learn an initiation classifier to be used during planning. To learn the rest of the model, we consider $f_\phi$ fixed and minimize the following losses and consider samples of the form $(s_{i-1}, o_{i-1}, s_i, o_i, r_i^\gamma, \tau_i)$ which can be obtained by slicing trajectories appropriately. We map them considering $f_\phi$ and minimize the following,

$$\mathcal{L}^R_\theta = (R_\theta(z_{i-1}, o_{i-1}, z_i, o_i) - r_i^\gamma)^2; \qquad \mathcal{L}^\tau_\theta = (\tau_\theta((z_{i-1}, o_{i-1}, z_i, o_i) - \tau_i)^2;$$

Finally, we minimize $\mathcal{L}_{\theta,\phi} = \beta_{\text{info}}\mathcal{L}_\phi + \beta_I\mathcal{L}^I_{\theta,\phi} + \beta_T\mathcal{L}^T_\theta + \beta_R\mathcal{L}^R_\theta + \beta_\tau\mathcal{L}^\tau_\theta$. In our experiments, all constants were $\beta_{\text{info}} = \beta_I = \beta_T = \beta_R = \beta_\tau = 1$.

### 4.3 Goal-based Planning with an Abstract Model

Consider a goal set $\mathcal{G} \subset S$ and $\mathcal{G}_\phi \subset \mathcal{Z}$, its mapping to $\mathcal{Z}$. In order to define the task MDP $M_\mathcal{G}$ for the agent to plan in, we define the task reward function for abstract state $\bar{s} = (\hat{z}, \hat{o}, z)$ as $R_\mathcal{G}(\bar{s}, o) = R_\theta(\bar{s}, o) + R_{\text{task}}\mathbb{1}[z \in \mathcal{G}_\phi]$ where $R_{\text{task}}$ is the goal reward. The first term can be interpreted as the base cost/reward of executing a skill while the second term indicates to the agent the task-specific rewarding states. Moreover, we augment the transition dynamics and set all $z \in \mathcal{G}_\phi$ as terminating states by setting $\bar{T}_\mathcal{G}(z_{\text{done}}|z, o) = \mathbb{1}[z \in \mathcal{G}_\phi]$. The agent uses the task MDP $\bar{M}_\mathcal{G}$ to simulate trajectories and improves its policy and it can, optionally, rollout the policy in the environment to collect new data that further improves the abstract model; this is shown in Algorithm 1.

## 5 Experiments

**Pinball environment (Konidaris and Barto, 2009)** This domain has a continuous state space with position vector $(x, y) \in [0, 1]^2$ and velocities $(\dot{x}, \dot{y}) \in [-1, 1]^2$. As opposed to its original formulation, we consider a variant with continuous actions that decrease or increase the velocity by $\Delta(\dot{x}, \dot{y}) \in [-1, 1]^2$. Moreover, we also consider the top view pixel observation of the environment as the agent's observation. As options, we handcrafted position controllers implemented as PID controllers that move the ball in the coordinate directions by a fixed step size.

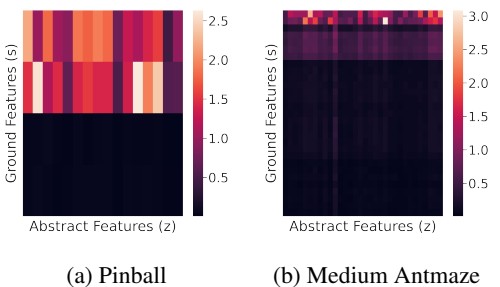

|        (a) Pinball        |   (b) Medium Antmaze   |

Figure 2: MI matrix: ground features $s$ are in the vertical axis and abstract features $z$ are in the horizontal axis. High MI (first two rows) correspond to position of the ball or the ant.

**Antmaze** We consider the problem of controlling a Mujoco (Todorov et al., 2012; Fu et al., 2020) Ant to navigate through a maze. The state space is a 29-dimensional vector that contains the position of the ant in the maze and the ant's proprioception. We consider the Medium Play maze as defined by (Fu et al., 2020). We use 8 options learned using TD3 (Fujimoto et al., 2018) that move the ant in the coordinate directions (north, south, east, west and the diagonal directions) in the maze by a fixed step size.

### 5.1 Abstract State Space Preserves Relevant Information for Planning

Our main hypothesis is that abstract actions drive state abstraction because the information needed to plan with a structured option set will be less than the ground perception space of the agent. To quantify this, we measure the information contained in the abstract state space about the ground features by estimating the MI using non-parametric methods based on $k$-nearest neighbors (Kozachenko and Leonenko, 1987). We use Scikit-learn implementation (Pedregosa et al., 2011). In Figure 2a, we show the MI matrix between Pinball's ideal features (position and velocities) and the learned features from the pixel observations. For Antmaze (Figure 2b), we purposely over-parameterized the abstract space to give enough capacity to learn the full observation, if necessary. However, we can see that features that are not necessary for planning with the skills are effectively abstracted away. In the case of Pinball only the first two dimensions corresponding to the ball position have high MI. In the Antmazes, similarly, the first 7 dimensions have the highest MI which corresponds to position in the maze (first two dimensions) and orientation of the ant's torso.

Qualitatively, we can visualize the learned abstract state space using Multidimensional Scaling (MDS) (Borg and Groenen, 2005). Figure 3 shows the abstract state space learned for the Antmaze and it reveals the pattern of the coordinate positions of the ant in the maze. Additionally, we show grounded observations that correspond to an abstract state: the ant at the represented position in the maze with many different configurations of the joints and torso.

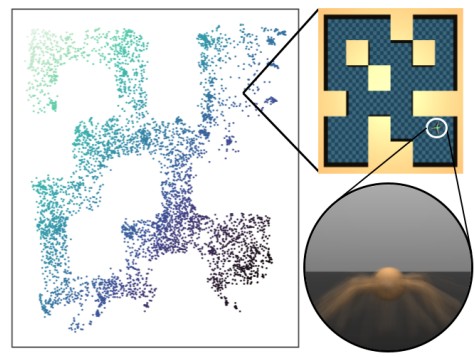

## 5.2 PLANNING WITH AN ABSTRACT MDP

We use our learned abstract models to plan using Double DQN (Van Hasselt et al., 2016). The DDQN agent rolls out trajectories in imagination to improve its policy and the new data collected is used to improve the model as in Algorithm 1. As our baseline, we use DDQN tuned to plan with the same options but interacting with the ground MDP.

Figure 3: **Medium Antmaze**. 2D MDS projection of the learned $\phi$: it learns to represent the position in the maze. The average grounding shows possible configurations of the ant joints when it is in the represented position .

For the pinball domain with use pixel observations. In Figure 4a, we compare learning curves averaged over 8 goals and 5 seeds where the gray area represent the number of samples required for pre-training the model. In Figure 4b, we show an analogous plot for the Antmaze (9 goals and 5 seeds). Both cases show the usefulness of the abstract models for diverse goal-based planning and the sample-complexity gain afforded by the model. Additionally, we show learning curves plot sequentially on Figure 4. The dashed lines separate learning curves for different goals. The bottom axis show the accumulated number of steps required by the ground agent and the top axis shows the steps needed by the agent using the abstract model. Notice that the top axis does not start at zero because of the initial samples required for model pre-training and we still observe a significant improvement in sample efficiency.

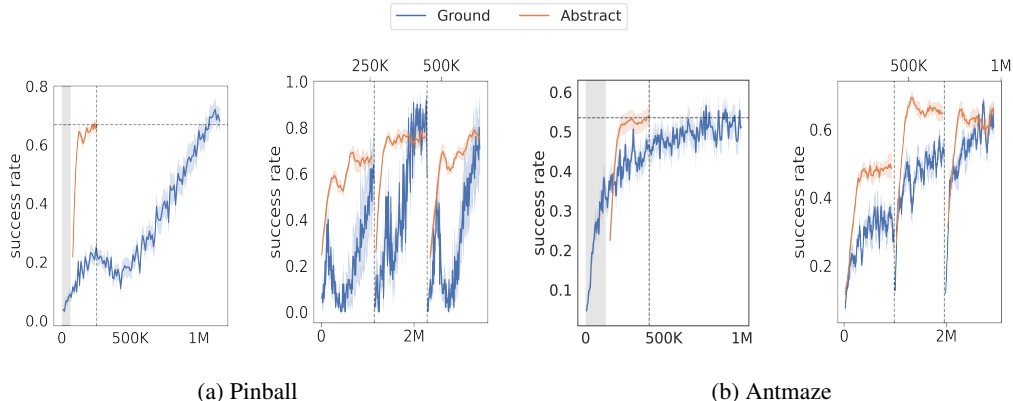

(a) Pinball

(b) Antmaze

Figure 4: Planning with an abstract model. (Left) Success rate v. Environment steps averaged over goals and 5 seeds. The gray area represents the offset for the steps needed to pre-train the model (Right) Success rate for a subset of the goals that shows the amount of steps needed for the ground environment (bottom axis) and the abstract agent (top axis).

## 6 RELATED WORKS

**Grounded Classical Planning** Konidaris et al. (2018) present a skill-driven method for constructing PDDL predicates (Fox and Long, 2003; Younes and Littman, 2004) for classical planning. This family of work formally bridges the options framework to classical planning, and recent work have extended this framework to work with portable skills (James et al., 2020) and object-centric skills (James et al., 2022), and to ground natural language in robotics (Gopalan et al., 2017). Importantly,

this framework offers guarantees that the learned grounded symbols support sound planning. A related body of work bridges deep learning with classical planning. Asai and Fukunaga (2018); Asai (2019); Asai et al. (2022) learn abstract binary representations to ground PDDL predicates and action operators from complex observations. Similarly, Ugur and Piater (2015a;b) approach the grounding problem by clustering action effects to create discrete symbols for planning, and Ahmetoglu et al. (2022) extends this approach to leverage deep learning methods. While these approaches manage to empirically work with complex observations, they do not offer formal guarantees that the symbols learned are sufficient for planning. Our approach, while not applied to classical planning, generalizes abstract state learning to continuous cases, it is compatible with the deep learning toolbox and it is theoretically principled.

**Model-based RL and State Abstractions** Learning MDP models from experience has been extensively studied (Sutton, 1991; Deisenroth and Rasmussen, 2011) for their benefits in generalization, sample efficiency, and knowledge transfer. Recent successful approaches use deep networks to handle complex observations spaces and long-term reasoning (Krishnan et al., 2015; Ha and Schmidhuber, 2018; Silver et al., 2018; Gregor et al., 2018; Buesing et al., 2018; Zhang et al., 2019; Hansen et al., 2022). An important challenge of this approach is learning an effective abstract state space and, most of them, have focused in learning abstract representations of complex observations based on reconstruction losses Gregor et al. (2018); Buesing et al. (2018); Zhang et al. (2019); Hafner et al. (2019; 2021; 2023). In contrast, recent approaches have moved away from this idea and focused in minimal abstract state spaces relevant for acting such as value prediction (Silver et al., 2018), Markov states (Gelada et al., 2019; Zhang et al., 2020; Allen et al., 2021; Nguyen et al., 2021), and controllability (Lamb et al., 2022). In fact, many of these explicitly use information maximization and information bottleneck approaches that are theoretically justified by our work.

From a theoretical point of view, there is extensive research to characterize the types of state abstractions (or state aggregation) (Li et al., 2006; Ferns et al., 2004; Castro and Precup, 2010) that are useful for RL. More recent work characterizes *approximate* state abstractions (Abel et al., 2016; 2018) that guarantee bounded value loss and the type of options that are compatible with a given state abstraction to guarantee value preservation (Abel et al., 2020).

**Temporally-extended Models** MDP models with skills have been recently considered in skill discovery research. Some work approach the problem assuming that the abstract state space is a graph and options are learned to reach the initiation set of another option (Bagaria and Konidaris, 2020; Bagaria et al., 2021a;b). Hafner et al. (2022) approaches the problem by building on the Dreamer algorithm (Hafner et al., 2019; 2021; 2023) and discover goals by abstracting over the learned abstract state. Similarly, Nair and Finn (2019) use generative models for subgoal generation and skill learning, and plan with a learned model in observation space. Other approaches learn forward dynamics models for skills discovered from an offline set of trajectories but do not abstract the state based on these skills (Freed et al., 2023; Shi et al., 2023; Zhang et al., 2023). While our method assumes that the options are given, it does not impose discrete constraints to the abstract state space, does not need to model the state dynamics at the finest time step, and it builds a principled abstract state space.

## 7 CONCLUSION

We introduce a method for learning abstract world models, designed to have agents with effective planning capabilities for goal-oriented tasks. Our core premise is that an agent must be capable of building a reusable abstract model for planning with a given skill set. We do this in a principled manner by characterizing the state abstraction that guarantees lossless planning in simulation. In other words, planning with a learned abstract model is sufficient to compute a policy for the real-world environment. Additionally, this work provides theoretical justification for information maximization approaches as a reliable strategy for learning abstract state representation learning.

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
