## A  APPENDIX

### A.1  PROOFS

**Theorem A.1.** *Let the tuple $(M, \bar{M}, G)$ be a grounded abstract model and a function $\phi : \mathcal{S} \to \mathcal{Z} \subseteq \mathbb{R}^{d_z}$. The model satisfies that $B_t(\cdot \mid o_0, ..., o_{t-1}) = \bar{B}_t(\cdot \mid o_0, ..., o_{t-1})$ if and only if $\phi$ is dynamics preserving.*

*Proof.* Let $\phi^{-1}(z) = \{s \in \mathcal{S} \mid \phi(s) = z\}$. We construct $\bar{T}$ and $G$ such that it satisfies that,

$$\bar{T}(z'|z, o) = \int_{s' \in \phi^{-1}(z')} T(s'|z, o)ds';$$

$$G(s'|z, o, z') = \frac{T(s'|z, o)\mathbb{1}[\phi(s') = z']}{\bar{T}(z'|z, o)}$$

If the dynamics preserving property holds, we have that there exists a mapping $\phi$ such that $T(s'|s, o) = T(s'|\phi(s), o)$. Hence, by defining that abstract state as $\bar{s} = (z, o, z')$, we can build the grounded abstract model such that it follows that $B_t = \bar{B}_t$, by construction.

To prove the converse, we assume that $B_t = \bar{B}_t$.

Hence, by construction, we have that $P(s_t, ..., s_0|o_0, z_0, ..., o_{t-1}, z_{t-1}) = \prod_t P(s_t|o_0, z_0, ..., o_{t-1}, z_{t-1})$. Therefore, we have that

$$\bar{B}_t(s_t, ..., s_0|o_0, ..., o_{t-1}) = \int \prod_{i=0}^{t} P(s_i|o_0, z_0, ..., o_{i-1}, z_{i-1})P(z_i, ..., z_0|o_0, ..., o_{i-1})dz_0...z_t$$

$$= \int \prod_{i=0}^{t} P(s_i|z_i, o_{i-1})P(z_i, ..., z_0|o_0, ..., o_{i-1})dz_0...z_t$$

$$= \int \prod_{i=0}^{t} G(s_i|z_{i-1}, o_{i-1}, z_i)P(z_i, ..., z_0|o_0, ..., o_{i-1})dz_0...z_t$$

$$= \prod_{i=0}^{t} \int G(s_i|z_{i-1}, o_{i-1}, z_i)P(z_i, z_{i-1}|o_0, ..., o_{i-1})dz_i z_{i-1}$$

$$= \prod_{i=0}^{t} \int G(s_i|z_{i-1}, o_{i-1}, z_i)\bar{T}(z_i|z_{i-1}, o_{i-1})P(z_{i-1}|o_0, ..., o_{i-2})dz_i z_{i-1}$$

$$= \prod_{i=0}^{t} \int \tilde{T}(s_i|z_{i-1}, o_{i-1})P(z_{i-1}|o_0, ..., o_{i-2})dz_{i-1}$$

$$B_t(s_t, ..., s_0|o_0, ..., o_{t-1}) = p_0(s_0) \prod_{i=1}^{t} T(s_i|s_{i-1}, o_{i-1})$$

$$= \prod_{i=1}^{t} T(s_i|s_{i-1}, o_{i-1})P(s_{i-1}|o_0, ..., o_{t-2})$$

Hence, we must have that for all $s_{i-1} \in z_{i-1}$ and all $i \in [t]$ and $t \geq 0$

$$\int T(s_i|s_{i-1}, o_{i-1})P(s_{i-1}|o_0, ..., o_{t-2})ds_{i-1} = \int \tilde{T}(s_i|z_{i-1}, o_{i-1})P(z_{i-1}|o_0, ..., o_{i-2})dz_{i-1}$$

That is,

$$\begin{cases} P(s_0) = p_0(s_0) = \int G(s|z_0)p_0(z_0)ds & \text{for } t = 0 \\ P(s_1|o_0) = \int T(s_1|s_0, o_0)p_0(s_0)ds_0 = \int \tilde{T}(s_0|z_0, o_0)p_0(z_0)dz_0 & \text{for } t = 1 \end{cases}$$

By definition, $t = 0$ holds. For $t = 1$, we have

$$
\begin{aligned}
P(s_1|o_0) &= \int T(s_1|s_0, o_0) p_0(s_0) ds_0 \\
&= \int T(s_1|s_0, o_0) G(s_0|z_0) p_0(z_0) dz_0 ds_0 \\
&= \int \tilde{T}(s_1|z_0, o_0) p_0(z_0) dz_0
\end{aligned}
$$

which follows from the equation at $t = 0$. Hence, it must be true that for any $s_0 \in \phi^{-1}(z_0)$, for any $z_0$ with $p_0(z_0) > 0$.

$$
\tilde{T}(s_1|z_0, o_0) = \int T(s_1|s_0, o_0) G(s_0|z_0) ds_0
$$

We can see that for any $s_0 \in \phi^{-1}(z_0)$ such that $T(s_1|s_0, o_0) \neq \tilde{T}(s_1|z_0, o_0)$, the abstract model would commit a non-zero error in its prediction. Hence, it must be that $T(s_1|s_0, o_0) = \tilde{T}(s_1|z_0, o_0)$ for $s_0 \in \phi^{-1}(z_0)$.

Let the equations at time $t = i - 1$ and $t = i - 2$ hold, then

$$
\begin{aligned}
P(s_i|o_0, ..., o_{i-1}) &= \int T(s_i|s_{i-1}, o_{i-1}) p_{i-1}(s_{i-1}|o_0, ... o_{i-2}) ds_{i-1} \\
&= \int T(s_i|s_{i-1}, o_{i-1}) \tilde{T}(s_{i-1}|z_{i-2}, o_{i-2}) p_{i-2}(z_{i-2}|o_0, ..., o_{i-3}) ds_{i-1} dz_{i-1} dz_{i-2} \\
&= \int T(s_i|s_{i-1}, o_{i-1}) G(s_{i-1}|z_{i-2}, o_{i-2}, z_{i-1}) \bar{T}(z_{i-1}|z_{i-2}, o_{i-2}) p_{i-2}(z_{i-2}|o_0, ..., o_{i-3}) ds_{i-1} dz_{i-1} dz_{i-2} \\
&= \int \tilde{T}(s_i|z_{i-1}, o_{i-1}) p_{i-1}(z_{i-1}|o_0, ..., o_{i-2}) dz_{i-1}
\end{aligned}
$$

Because $p_{i-1}(z_{i-1}|o_0, ..., o_{i-2}) = \int \bar{T}(z_{i-1}|z_{i-2}, o_{i-2}) p_{i-2}(z_{i-2}|o_0, ..., o_{i-3}) dz_{i-2}$ hold by construction of the abstract MDP, we need the following to hold.

$$
\tilde{T}(s_i|z_{i-1}, o_{i-1}) = \int T(s_i|s_{i-1}, o_{i-1}) G(s_{i-1}|z_{i-2}, o_{i-2}, z_{i-1}) ds_{i-1}. \tag{4}
$$

Therefore, as in the base case, we need that $\tilde{T}(s_i|z_{i-1}, o_{i-1}) = T(s_i|s_{i-1}, o_{i-1})$ for all $s_{i-1} \in \phi^{-1}(z_{i-1})$ that have $G(s_{i-1}|z_{i-2}, o_{i-2}, z_{i-1}) > 0$. Then, $\phi$ must be dynamics preserving.

$\square$

**Corollary A.2.** *Let the tuple $(M, \bar{M}, G)$ be a grounded abstract model. Let the strong subgoal property (Konidaris et al., 2018) for an option o be defined as, $Pr(s'|s, o) = Pr(s'|o)$. The dynamics preserving property holds with a finite abstract state space $\mathcal{Z} = [N]$ for some $N \in \mathbb{N}$ if and only if the strong subgoal property holds.*

*Proof.* If the strong subgoal property holds, we have that $Pr(s'|s, o) = Pr(s'|o)$. Then, for any function $\phi : S \to \mathcal{Z}$, it holds that $P(s'|\phi(s), o) = P(s'|s, o)$.

Therefore, it is only important to be able to know if a given option is executable in a given abstract state. Therefore, we can construct the function $I_{\mathcal{O}}(s) = [I_0(s), ..., I_{|\mathcal{O}|}(s)]$ that returns a binary vector that indicates which options are executable in $s$.

Define the equivalence relation $s_0 \sim_{\mathcal{O}} s_1$ iff $I_{\mathcal{O}}(s_1) = I_{\mathcal{O}}(s_2)$. We can define the abstract state space as $Z \triangleq S/\sim_{\mathcal{O}}$, that is, the set of equivalent classes. Given that there at most $2^{|\mathcal{O}|} \in \mathbb{N}$ classes, then the abstract MDP is finite.

We assume that the dynamics preserving property holds and that the abstract state space $Z$ is finite to prove the converse. Then, there exists $\phi : S \rightarrow \mathcal{Z}$ such that $P(s'|\phi(s), o) = P(s'|s, o)$ and $P(I_o = 1|s) = P(I_o = 1|\phi(s))$.

We can construct a factored $\phi(s) = [\phi_D(s), \phi_I(s)]$, such that, $P(s'|\phi(s), o) = P(s'|\phi_D(s), o)$ and $P(I_o = 1|\phi(s)) = P(I_o = 1|\phi_I(s))$.

If we define $\phi_I$ based on the function $I_\mathcal{O}$, as before, then $\phi_I$ maps to a set of at most $2^{|\mathcal{O}|}$ elements. As $\mathcal{Z} = \mathcal{Z}_D \times \mathcal{Z}_I$ is finite, then $\mathcal{Z}_D$ is also finite. Thus, we construct $Z_D = [M]$ and for each option $o$ and equivalence class $m \in [M]$ options from each option $o$ such that $Pr(s'|o_m) \triangleq Pr(s'|m, o)$. Then, the strong subgoal property holds for every $o_m$.

$\square$

**Proposition A.3.** *Let $\phi$ be a dynamics-preserving abstraction and $\bar{s} = (\hat{z}, \hat{o}, z)$. For $\epsilon > 0$, if $\|G_z(s) - G_{\bar{s}}(s)\|_1^2 \leq \epsilon$, then there exists $\epsilon_T > 0$ and $\epsilon_R > 0$ such that $\|T(s'|s, o) - \tilde{T}(s'|z, o)\|_1^2 \leq \epsilon_T$ and $\|R(s, o) - \tilde{R}(z, o)\|_1^2 \leq \epsilon_R$.*

*Proof.* First, we prove that the bounded grounding error implies bounded transition distribution error. If $\phi$ is a dynamics abstraction, then we can learn $\tilde{T}(z'|z, o)$ and we have that $T(s'|s, o) = T(s'|z, o) = \int G_{\bar{s}}(s)\bar{T}(z'|z, o)dz'$ and its corresponding approximation $\tilde{T}(s'|z, o) = \int G_{z'}(s)\bar{T}(z'|z, o)dz'$

$$
\begin{aligned}
\|T(s'|s, o) - \tilde{T}(s'|z, o)\|_1 &= \left| \int \left( G'_{\bar{s}}(s)\bar{T}(z'|z, o) - G_{z'}(s)\bar{T}(z'|z, o) \right) dz' \right| \\
&\leq \int \bar{T}(z'|z, o)|G_{\bar{s}'}(s) - G_{z'}(s)|dz'ds \\
&\leq \sqrt{\epsilon}
\end{aligned}
$$

Analogously, we can bound the error of the reward function.

$$
\begin{aligned}
\|\bar{R}(z', o) - \tilde{R}(z', o)\|_1 &= \left| \int G_{\bar{s}'}(s)R(s, o)ds - \int G_{z'}(s)R(s, o)ds \right| \\
&\leq \int |G_{\bar{s}'}(s) - G_{z'}(s)| \, |R(s, o)| \, ds \\
&\leq RMax \int |G_{\bar{s}'}(s) - G_{z'}(s)| \, ds \\
&\leq RMax\sqrt{\epsilon}
\end{aligned}
$$

Then, it follows from Minkowski's inequality that

$$
\begin{aligned}
\|R(s, o) - \bar{R}(z', o)\|_1 &= \|R(s, o) - \tilde{R}(z', o) + \tilde{R}(z', o) - \bar{R}(z', o)\|_1 \\
&\leq \|R(s, o) - \tilde{R}(z', o)\|_1 + \|\tilde{R}(z', o) - \bar{R}(z', o)\|_1 \\
&\leq \sqrt{\epsilon} + RMax\sqrt{\epsilon} = \sqrt{\epsilon_R}
\end{aligned}
$$

$\square$

**Theorem A.4** (Value Loss Bound). *Let $(M, \bar{M}, G)$ be a grounded abstract model and $\tilde{T}(s'|\bar{s}, o) = \int G_{\bar{s}'}(s')\bar{T}(\bar{s}'|\bar{s}, o)d\bar{s}'$ be the approximate transition dynamics from the grounded model. If the following conditions hold for all $o \in \mathcal{O}$ and all $s \in \mathcal{S}$ with $G_{\bar{s}}(s) > 0$: (1) $\|T(s'|s, o) - \tilde{T}(s'|\bar{s}, o)\|_1^2 \leq \epsilon_T$, and (2) $|R(s, o) - \bar{R}(\bar{s}, o)|^2 \leq \epsilon_R$; then, for any policy $\pi$,*

$$
|Q^\pi(s, o) - Q^\pi(\bar{s}, o)| \leq \frac{\sqrt{\epsilon_R} + \gamma VMax\sqrt{\epsilon_T}}{1 - \gamma}.
$$

*Proof.* We proceed by induction on $Q_n^\pi(\bar{s}, o)$, where

$$v_0^\pi(\bar{s}) = \mathbb{E}_{s \sim \bar{s}}\left[v^\pi(s)\right], \tag{5}$$

$$Q_1^\pi(\bar{s}, o) = \int_{s \in \mathcal{S}} P(s)\left(R(s, o) + \gamma^\tau v_0^\pi(\bar{s}')\right)ds, \tag{6}$$

$$= \int_{s \in \mathcal{S}} P(s)\left(R(s, o) + \gamma^\tau \int_{s' \in \mathcal{S}} T^{s,o,s'} v^\pi(s')ds'\right)ds, \tag{7}$$

$$Q_i^\pi(\bar{s}, o) = \int_{s \in \mathcal{S}} P(s)\left(R(s, o) + \gamma^\tau v_{i-1}^\pi(\bar{s}')\right)ds, \tag{8}$$

with $\bar{s}' = T(\cdot \mid s, o)$. I use $P(s)$ as shorthand for $P(s \sim \bar{s})$ and $T^{s,o,s'}$ for $T(s' \mid s, o)$, and let

$$\epsilon_{Q,n} = \sum_{i=0}^n \sqrt{\epsilon_R} + \gamma^i \left(\text{VMAX}\sqrt{\epsilon_T}\right). \tag{9}$$

*Base Case:* $Q^\pi \approx Q_1^\pi$.

$$Q^\pi(s, o) - Q_1^\pi(\bar{s}, o) \tag{10}$$

$$= R(s, o) + \gamma^\tau \int_{s'} T^{s,o,s'} v^\pi(s')ds' - \int_s P(s)\left(R(\bar{s}, o) - \gamma^\tau v_0^\pi(\bar{s}')ds\right), \tag{11}$$

$$= \underbrace{R(s, o) - R(\bar{s}, o)}_{\leq \sqrt{\epsilon_R}} + \gamma^\tau \int_{s'} T^{s,o,s'} v^\pi(s')ds' - \int_s P(s)\gamma^\tau v_0^\pi(\bar{s}')ds, \tag{12}$$

$$\leq \sqrt{\epsilon_R} + \gamma^\tau \int_{s'} T^{s,o,s'} v^\pi(s')ds' - \gamma^\tau \int_s P(s)_{s' \sim \bar{s}'}[v^\pi(s')]ds \tag{13}$$

$$\leq \sqrt{\epsilon_R} + \gamma^\tau \int_{s'} T^{s,o,s'} v^\pi(s')ds' - \gamma^\tau \int_s P(s) \int_{s'} P(s' \sim \bar{s}')v^\pi(s')ds' \, ds, \tag{14}$$

$$\leq \sqrt{\epsilon_R} + \gamma^\tau \int_{s'} T^{s,o,s'} v^\pi(s')ds' - \gamma^\tau \int_s P(s) \int_{s'} T^{s,o,s'} v^\pi(s')ds' \, ds, \tag{15}$$

$$\leq \sqrt{\epsilon_R} + \gamma^\tau \text{VMAX} \underbrace{\int_{s'} T^{s,o,s'} - \int_s P(s)T^{s,o,s'}ds \, ds'}_{\leq \sqrt{\epsilon_T}}, \tag{16}$$

$$\leq \sqrt{\epsilon_R} + \gamma^\tau \text{VMAX}\sqrt{\epsilon_T}. \tag{17}$$

This concludes the base case. $\qquad\square$

*Inductive Case:* $Q^\pi \approx Q_n^\pi \implies Q^\pi \approx Q_{n+1}^\pi$. We assume that, for every $s \in \mathcal{S}$ and any $o$,

$$Q^\pi(s, o) - Q_n^\pi(\bar{s}, o) \leq \epsilon_{Q,n}, \tag{18}$$

and prove that

$$Q^\pi(s, o) - Q_{n+1}^*(\bar{s}, o) \leq \epsilon_{Q,n+1}. \tag{19}$$

By algebra,

$$Q^\pi(s, o) - Q_{n+1}^\pi(\bar{s}, o) \tag{20}$$

$$= R(s, o) + \gamma^\tau \int_{s'} T^{s,o,s'} v^\pi(s')ds' - \int_s P(s)\left(R(s, o) + \gamma^\tau v_n^\pi(\bar{s}')\right)ds, \tag{21}$$

$$= \underbrace{R(s, o) - R(\bar{s}, o)}_{\leq \sqrt{\epsilon_R}} + \gamma^\tau \int_{s'} T^{s,o,s'} v^\pi(s')ds' - \gamma^\tau \int_s P(s)v_n^\pi(\bar{s}')ds, \tag{22}$$

$$\leq \sqrt{\epsilon_R} + \gamma^\tau \int_{s'} T^{s,o,s'} v^\pi(s')ds' - \gamma^\tau \int_s P(s)v_n^\pi(\bar{s}')ds, \tag{23}$$

$$= \sqrt{\epsilon_R} + \gamma^\tau \int_{s'} T^{s,o,s'} v^\pi(s')ds' - \gamma^\tau \int_s P(s) \underbrace{v_n^\pi(\bar{s}')}_{\geq_{s' \sim \bar{s}'}[v^\pi(s')] - \epsilon_{Q,n}} ds, \tag{24}$$

$$\leq \sqrt{\epsilon_R} + \gamma^\tau \int_{s'} T^{s,o,s'} v^\pi(s')ds' - \gamma^\tau \int_s P(s) \left( {}_{s' \sim \bar{s}'}[v^\pi(s')] - \epsilon_{Q,n} \right) ds, \tag{25}$$

$$= \sqrt{\epsilon_R} + \gamma^\tau \int_{s'} T^{s,o,s'} v^\pi(s')ds' - \gamma^\tau \int_s P(s) \int_{s'} T^{s,o,s'} v^\pi(s')ds' \, ds + \gamma^\tau \epsilon_{Q,n}, \tag{26}$$

$$= \sqrt{\epsilon_R} + \gamma^\tau \int_{s'} T^{s,o,s'} v^\pi(s')ds' - \gamma^\tau \int_{s'} \underbrace{\int_s P(s)T^{s,o,s'} v^\pi(s')ds}_{=T^{\bar{s},o,s'}} ds' + \gamma^\tau \epsilon_{Q,n}, \tag{27}$$

$$\leq \sqrt{\epsilon_R} + \gamma^\tau \text{VMAX} \underbrace{\int_{s'} T^{s,o,s'} - T^{\bar{s},o,s'} ds'}_{\leq \sqrt{\epsilon_T}} + \gamma^\tau \epsilon_{Q,n}, \tag{28}$$

$$\leq \sqrt{\epsilon_R} + \gamma^\tau \text{VMAX} \sqrt{\epsilon_T} + \gamma^\tau \epsilon_{Q,n}, \tag{29}$$

$$\leq \sqrt{\epsilon_R} + \gamma \text{VMAX} \sqrt{\epsilon_T} + \gamma \epsilon_{Q,n}, \tag{30}$$

$$= \epsilon_{Q,n+1}. \tag{31}$$

This concludes the inductive case. $\square$

Thus, by induction and the convergence of the geometric series, for any $s, o, \pi$, we conclude that

$$Q^\pi(s,o) - Q^\pi(\bar{s},o) \leq \frac{\sqrt{\epsilon_R} + \gamma \text{VMAX} \sqrt{\epsilon_T}}{1-\gamma}. \tag{32}$$

$\square$

### A.2 TPC IS DYNAMICS PRESERVING

We start by considering that by learning an abstract state space such that $MI(S'; Z, O)$ is maximized. The following decomposition based on the mutual information chain rule corresponds to the TPC algorithm (Nguyen et al., 2021). In the original paper, they work at the primitive action level and all actions available always, hence, there's no need to consider initiation sets.

$$MI(S', Z'; Z, O) \overset{(a)}{=} MI(S'; Z, O) + \underbrace{MI(Z'; Z, O|S')}_{=0};$$

$$\overset{(b)}{=} MI(Z'; Z, O) + \underbrace{MI(S'; Z, O|Z')}_{(1)};$$

$$\overset{(c)}{=} MI(Z'; Z, O) + MI(S'; Z, A) - MI(S'; Z') + MI(S'; Z'|Z, O);$$

where (a) follows from the fact that give $s'$ we can determine $z'$, (b) follows from decomposing the term on the left-hand size and (c) from decomposing term (1).

The above implies that $MI(Z'; Z, O) = MI(S'; Z') - MI(S'; Z'|Z, O)$. Therefore, if we maximize both sides of this identity, we must have a latent space that preserve *only* the information of the state $s'$ that is predictable from the previous $(z, a)$ pair. $MI(Z'; Z, O)$ ensures that the next abstract state is predictable from the $(z, o)$ tuple. $MI(S; Z)$ ensures that the abstract state has information about the ground state which is measured by $g(s|z)$.

$$MI(S; O) = \int p(s, z) \log \frac{g(s|z)}{p(s)} ds dz \tag{33}$$

The following decomposition shows the two extra terms required by the TPC algorithm to estabilize the optimization. Term $(a)$ is the (differential) entropy of $\phi$ which tends to infinity for a deterministic function. This is solved by smoothing it with Gaussian noise of 0 mean and fixed standard deviation, as done in TPC. The second term $(b)$ corresponds to the consistency term, that is, the transition function $p(z'|z, a)$ must have low entropy, which ensures that the abstract dynamics are learnt.

$$
\begin{aligned}
M(S'; Z'|Z, O) &= \int p(s', z', z, o) \log \frac{p(s', z'|z, o)}{p(s'|z, o)p(z'|z, o)} ds' dz' dz do \\
&= \int p(s', z', z, o) \log \frac{p(z'|s')}{p(z'|z, o)} \\
&= \underbrace{\int p(s', z') \log p(z'|s') ds' dz'}_{(a)} - \underbrace{\int p(z', z, o) \log p(z'|z, o) dz' dz do}_{(b)}
\end{aligned}
$$

By maximizing $MI(Z'; Z, O)$ and $MI(S'; Z')$ using InfoNCE (Oord et al., 2018), we obtain the TPC algorithm.

## B  Experiments

For all our planning experiments we use DDQN (Van Hasselt et al., 2016) modified to consider initiation sets for action selection and target computation to make it compatible with options. We use Adam (Kingma and Ba, 2014) as optimizer. As exploration, we use linearly decaying $\epsilon$-greedy exploration.

### B.1  Experiments

#### B.1.1  Environments

**Pinball Domain (Konidaris and Barto, 2009)**    We use a continuous action variant of the original environment. The state space $s = (x, y, \dot{x}, \dot{y})$ with $(x, y) \in [0, 1]^2$ and $(\dot{x}, \dot{y}) \in [-1, 1]$. The action space is the ball acceleration expressed in the form of $\Delta(\dot{x}, \dot{y} \in [-1, 1]^2$. The layout of the obstacles is as in the original environment, show in Figure 8. The reward function takes $-5$ per unit of acceleration. The discount factor is $\gamma = 0.9997$.

**Pinball Options**    Pinball options were designed to the agent in the coordinate dimensions by step size $0.04$. The initiation set are all the position in which the ball would not hit an obstacle by moving in the desired direction. The termination probability is determined by a Gaussian centered in the goal position with standard deviation as $0.01$. For the policy, we handcrafted PI controllers for the position with constants $K_p = 50$ and $K_i = 8$.

**Antmazes**    We consider the U-Maze and Medium-Play mazes implemented by D4RL (Fu et al., 2020) with the Mujoco ant. In Figure 9 we show diagrams of the considered mazes. The state space is $\mathcal{S} \in \mathbb{R}^{29}$, where the first two dimensions corresponds to the position of the ant in the maze and the rest is proprioception for the ant controls. The action space is $\mathcal{A} \subset [-1, 1]^8$ to control the ant joints.

**Antmaze Options**    We consider options that move the ant in the 8 directions (North, South, East, West, North-East, North-West, South-East, South-West) by a distance of $1$ unit. For the position controller, we train a goal-conditioned policy using HER (Andrychowicz et al., 2017) and TD3 (Fujimoto et al., 2018) that would take a goal position in an drive walk the ant to it. This is generally hard for arbitrary goals given the separation between the current position and the goal, however, we only needed the policy to become accurate for short distances, so we sampled initial positions within $1.5$ of the desired goal. The goals were sampled uniformly over the possible positions in the maze. Then we learned the initiation sets as classifiers were the option execution would be successful. The termination condition is a threshold of $0.5$ distance to the goal.

#### B.1.2  Network Architectures

**Pixel Observations**    As encoder for pixel observation, we use ResNet Convolutional Networks, as used in Dreamer (Hafner et al., 2021). The ResNet starts with an initial $24$ depth and doubles in depth until reaching the minimal resolution. See Table 1.

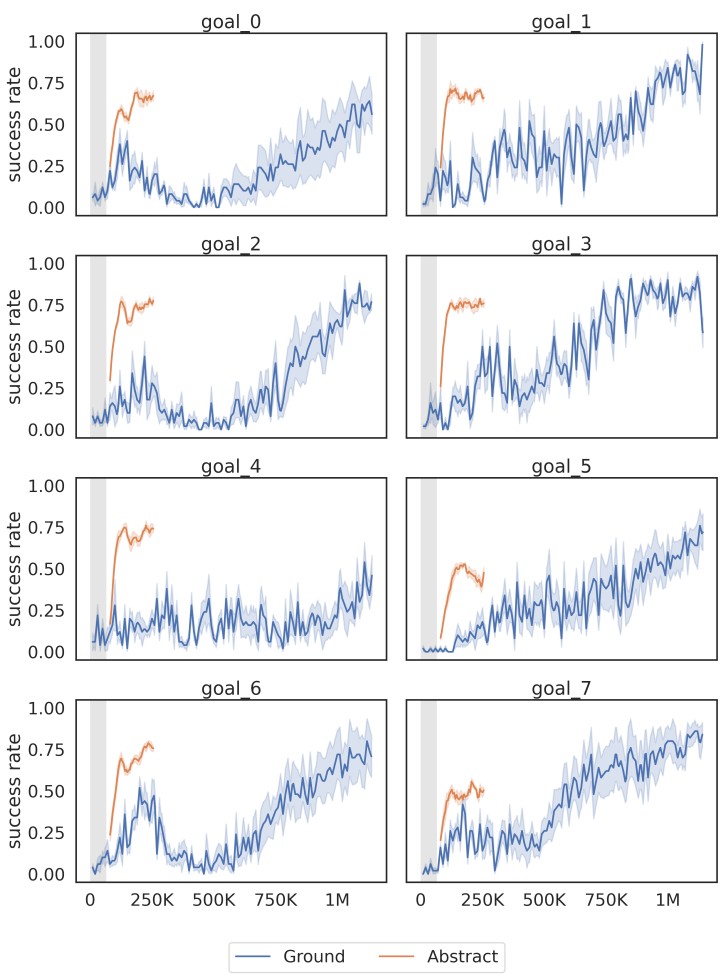

Figure 5: **Pinball from pixels**. Ground baseline vs Abstract planning. Each goal learning curve is averaged over 5 seeds and 1 standard deviation shown in the shaded area of each curve. The gray area corresponds to the offset that corresponds to samples used to pre-train the model. Although is shown in every plot, it is common to all goals.

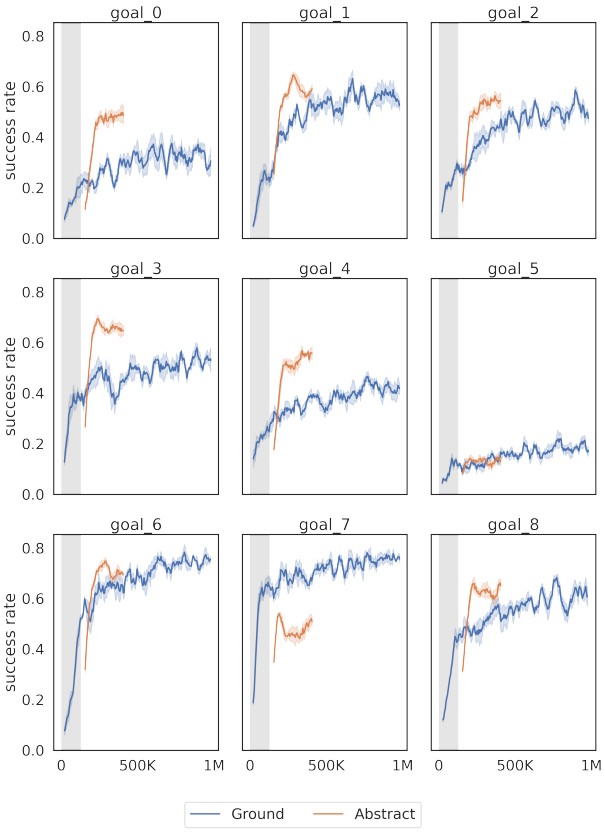

Figure 6: **Medium Play Antmaze**. Ground baseline vs Abstract planning. Each goal learning curve is averaged over 5 seeds and 1 standard deviation shown in the shaded area of each curve. The gray area corresponds to the offset that corresponds to samples used to pre-train the model. Although is shown in every plot, it is common to all goals.

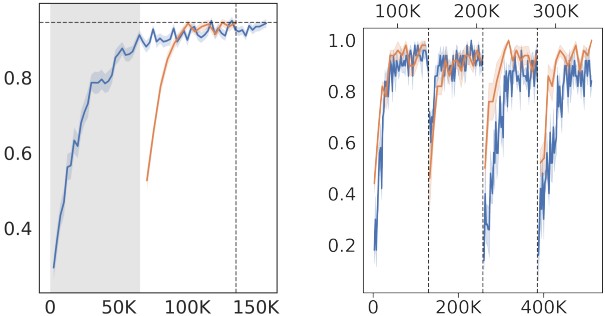

Figure 7: **U-Maze Antmaze**.

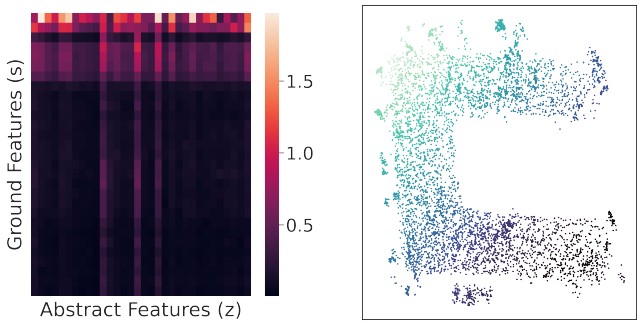

Figure 8: **U-Maze Antmaze**.

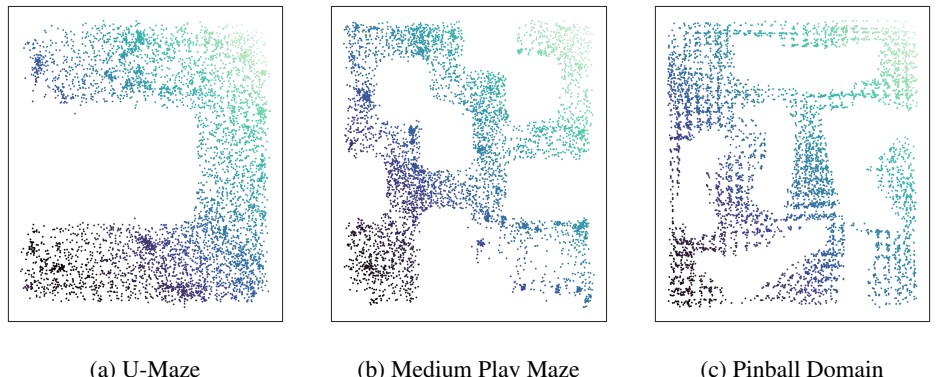

(a) U-Maze  (b) Medium Play Maze  (c) Pinball Domain

Figure 9: Ground truth visualization of possible positions of the agent in the evaluation Environments

Table 1: ResNet CNN Configuration

| Parameter | Value |
|-----------|-------|
| in width | 50 |
| in height | 50 |
| color channels | 1 |
| depth | 24 |
| cnn blocks | 2 |
| min resolution | 4 |
| mlp layers | $[256,]$ |
| outdim | 4 |
| mlp activation | silu |
| cnn activation | silu |

**MLP Architectures**  For all other models, we use MLPs with the relevant input and output dimensions. This includes encoder, initiation classifiers, transition function, reward function and duration. For the reward function we use the symlog transformation (Hafner et al., 2021) and a log transformation for the option duration network.

Table 2: MLP Configuration

| Parameter | Value |
|-----------|-------|
| hidden dims | $[128, 128]$ |
| activation | relu |

**Density Estimation**  We use mixture of Gaussians with 4 components and Gaussians with diagonal covariance matrices. We use the reparameterization trick (Kingma and Welling, 2013) to optimize the mean and variance functions.

### B.1.3  AGENT HYPERPARAMETERS

To train our baseline DDQN agent with the following parameters that we tune by doing grid search for 5 goal positions and 2 seeds, we use all these parameters to learn for all goals.

**Pinball Domain**  For pixel observations we use the same architecture as described before for the world model encoder. For simpler observation, we use an MLP as before.

Table 3: Pinball ground DDQN parameters

| Parameter | Value |
|-----------|-------|
| final exploration steps | 500000 |
| final epsilon | 0.1 |
| eval epsilon | 0.001 |
| replay start size | 10000 |
| replay buffer size | 500000 |
| target update interval | 10000 |
| steps | 1250000 |
| update interval | 5 |
| num step return | 1 |
| learning rate | $10^{-5}$ |
| $\gamma$ | 0.9997 |

### B.1.4  WORLD MODEL HYPERPARAMETERS

Table 4: Ground DDQN Parameters for the Antmazes

| Parameter | Value |
|---|---|
| final exploration steps | $350,000$ |
| final epsilon | $0.1$ |
| eval epsilon | $0.001$ |
| replay start size | $1,000$ |
| replay buffer size | $100,000$ |
| target update interval | $1,000$ |
| steps | $1,000,000$ |
| update interval | $5$ |
| num step return | $1$ |
| learning rate | $5 \times 10^{-4}$ |
| $\gamma$ | $0.995$ |

Table 5: U-Maze Imagination DDQN Parameters

| Parameter | Value |
|---|---|
| final exploration steps (proportion) | 30% of agent training steps |
| final epsilon | 0.1 |
| eval epsilon | 0.001 |
| replay start size | 1000 |
| replay buffer size | 100000 |
| target update interval | 10000 |
| update interval | 5 |
| num step return | 1 |
| learning rate | $1 \times 10^{-4}$ |
| rollout length | 100 |

Table 6: World Model Parameters

| Parameter | Value |
|---|---|
| buffer size | $100,000$ |
| batch size | 16 |
| learning rate | $1 \times 10^{-4}$ |
| train every | 8 |
| max rollout length | 64 |