# OpenReview forum: "Learning Abstract World Models for Value-preserving Planning with Options"
_ICLR.cc/2024/Conference — Submitted to ICLR 2024_

### Official Review · Reviewer_NRqK · 2023-10-19

**Soundness:** 3 good
**Presentation:** 2 fair
**Contribution:** 2 fair
**Rating:** 3
**Confidence:** 4

**Summary:**

This paper proposes an algorithm for learning MDP state abstractions that preserve information needed for planning (namely, the values of states). A major differentiator from symbolic approaches is the idea that these state abstractions should be continuous rather than discrete. The key assumption is that you are given a set of options and a dataset obtained by rolling them out. Experiments are conducted in a few simple domains: pinball and antmaze, and demonstrate that the learned abstractions are sensible.

**Strengths:**

The paper addresses an important topic (abstraction learning) and I appreciate the theoretically motivated algorithms. This line of work is of great interest to many attendees of ICLR. I also appreciate that the authors were clear about wanting continuous representations right off-the-bat. The math is also correct as far as I was able to tell, though I didn't check the proofs in the appendix in careful detail.

**Weaknesses:**

Unfortunately, I recommend rejection for this paper due to 4 major reasons: 1) unconvincing experiments, 2) missing key citations to related work, 3) issues in technical details, and 4) unclear motivation.

1) unconvincing experiments

The experiments in this paper are very basic and only serve as a simple proof-of-concept that the learned abstractions are somewhat useful. To really scale up the experiments to the level expected for a conference paper, I would expect to see evidence that the learned abstractions are useful in more hierarchical domains (e.g., classic domains from the options literature like keys and doors). In such domains, we could test whether the value-preserving property holds empirically, by comparing the values from planning under the abstract model to the (ground truth) values from planning under the true model.

Additionally, I would like to see comparisons to many more RL algorithms, especially hierarchical ones like HIRO (https://arxiv.org/abs/1805.08296), HVF (https://arxiv.org/abs/1909.05829), and Director (https://arxiv.org/abs/2206.04114). This is because at the end of the day, the authors are proposing to learn a state encoder $\phi$, and despite all the theory that has gone into their algorithm, the question that must be answered is whether this $\phi$ outperforms the encoders learned by all these other SOTA hierarchical RL algorithms.

2) missing key citations to related work

The authors are missing several key citations, the most important of which is the line of work by David Abel, such as "Near optimal behavior via approximate state abstraction" (https://proceedings.mlr.press/v48/abel16.html) and "Value preserving state-action abstractions" (https://proceedings.mlr.press/v108/abel20a/abel20a.pdf). Those papers have very similar theory to what appears in this one, and so the novelty of the proposed approach is unclear. There are also less-famous but still important-to-cite papers from other authors, like "Abstract value iteration for hierarchical reinforcement learning" (https://proceedings.mlr.press/v130/jothimurugan21a/jothimurugan21a.pdf) and "Deciding what to model: Value-equivalent sampling for reinforcement learning" (https://proceedings.neurips.cc/paper_files/paper/2022/hash/3b18d368150474ac6fc9bb665d3eb3da-Abstract-Conference.html). It is important for the authors to contextualize the contributions of this paper against all these related works.

3) issues in technical details

The authors say in Section 3.2 that when B = \bar{B}, "then simulating a trajectory in the abstract model is the same as in the ground model". But I don't think this is true, because we need the rewards to match between the two trajectories too, and $B_t$ says nothing about rewards, only dynamics. The authors go on to say: "Therefore, planning in the abstract model is accurate, in the sense, that the value of an abstract state z computed in the abstract model is the same as the one would get from trajectories from the ground MDP for the abstraction operator G." Again, I think this is wrong because it ignores the abstract reward function, which could be arbitrarily different from the ground one. In fact, in the proof of corollary 3.8, the authors assume $E_{s \sim G(\cdot \mid z)}[R(s, o)] = \bar{R}(z, o)$, and it's only _under this assumption_ that the claims hold. But combining this assumption on reward function with Definition 3.6 ends us back up at the bisimulation conditions, and then it's not clear what the contributions of this paper are.

As a separate point, the second term in the mutual information expression of Section 4.2, $MI(S'; Z, A)$, seems very extreme! It is saying that you have to be able to predict the entire ground next state from the current abstract state and action. Doesn't this means the abstraction can't lose any information? This seems like an important technical limitation of the approach.

4) unclear motivation

The authors often state that a discrete abstract state space is bad, when pointing to work on symbolic abstraction learning (e.g., PDDL). But it's not clear why this is really bad. The authors say discrete abstract states are "not applicable when planning with the available high-level actions requires a continuous state representation", but this doesn't make sense to me, as the options have to act in the ground environment states, not in the abstract state space, and so the options could be defined with respect to either a discrete or a continuous abstract state space. Furthermore, it can be much easier to plan in a discrete abstraction (e.g., using powerful symbolic planners).

I believe a fruitful research direction would be to compare the abstractions learned by a symbolic approach against the abstractions learned by a continuous approach (like the authors').

**Questions:**

Questions:
* Not much is said about the dataset $\mathcal{D}$, but intuitively, it has to be "good" in order for the learned state abstraction to be reasonable. In particular, the agent must see all the options being executed in a variety of settings, and obtain good coverage over the state-action space. Are there any concrete statements we can make about what properties we need this dataset to have?
* "we must build a model of its effect" Do you mean to say "of the effect of each option"?
* "with mean value equal to that by planning with the original MDP" What is the mean over?
* Why did we switch from using O (denoting the option set) everywhere to using A throughout Section 4? Shouldn't we continue to use O, unless I am misunderstanding something?
* Section 4.3: Why should there be any cost/reward associated with executing skills? Shouldn't a sparse reward for reaching the goal be enough?
* Eq 2: What are the "I" random variables inside the mutual information expression referring to?

Minor edits:
* "make the same decision" To clarify, we just need that the policy maps all states in z to the same action distribution. A stochastic policy isn't really committing to a "decision" about what action to take.
* "Abstractions alleviate this tension: action abstractions enable agents to plan at larger temporal scales and state abstractions reduce the complexity of learning and planning" I would say that both of them do both of these. Action abstractions certainly reduce the complexity of planning, which is typically exponential in the branching factor.
* "learns a further abstraction" --> "learn a further abstraction"
* "otherwise it is referred as learning" I would say "policy learning" to distinguish from other things you might learn
* "when it is the given position" --> "when it is in the given position"
* "referred as learning" --> "referred to as learning"
* "results a bounded value loss" --> "results in a bounded value loss"
* In definition 3.5, the authors use $s_o$ in a few places where they mean $s_0$.

---

> ### Author Response · Authors · 2023-11-16
>
> - **Unconvincing experiments**: We are currently working to add MBRL baselines to our experiments. However, we think it's important to notice that HIRO and Director do *skill discovery*. We assume that skills are given and we leave for future work to use our framework in a loop of skill discovery.
>
> - **Missing references**: Thank you for pointing out these references. We have now included a new paragraph in related works to include classical work in state aggregation theory in RL---that is definitely relevant to our work. However, it is worth pointing out that our proposed framework builds over probabilistic groundings that are more general than state aggregation: instead of partitioning the state space, we can have overlapping supports for abstract states. Therefore, a ground state might belong to more that one abstract state.
>
> - **Issues in technical details**:
> We understand the concern on the relationship between dynamics-preserving abstractions and bisimulation (and similar), and we have updated our writing to help with this.
>
> - However, we argue that the purpose of an abstract model for a set of skills is to be able to solve many new tasks with those skills by planning. Hence, the ground reward *is not meant to represent a task* but to represent the cost or reward associated with executing skills (e.g. energy cost to execute a particular motor skill).
> Therefore, we are not interested in preserving Markovinity with respect to the ground reward because there’s a limit to how much we can abstract away then. For instance, in the Antmaze experiment, if the ground reward is dependent on the joints' states then we wouldn't be able to abstract beyond the expert features. Therefore, we estimate the average cost of executing an option when it start at $s$ and ends in $s'$ (averaged over possible trajectories---see ground reward definition).
> When we move to the abstract level, given an abstract state we don't know exactly in what ground state we would be, hence, the best we can do is to estimate the average ground reward under the grounding function, and in turn **we preserve the mean value with respect to the grounding function**.
>
> We thank you for pointing this out and we have updated our writing to reflect this.
>
> - **The dynamics preserving is, in fact, strong**. However, this is similar to what current MBRL methods use to build models with latent variables. It means that $\phi(s)$ has to be a sufficient statistic to predict the next state (e.g., reconstruction based representation learning). In the worst case, in which everything in $s$ is necessary to predict the next state $s'$, then we would expect $\phi$ to reduce to the identity. However, with a structured set of options, we expect that some features of the state are controlled and others won't be (e.g. ant's position in the maze). Therefore, because we do not know which state trajectory was taken by the option to get from $s\rightarrow s'$, then some details of the state will become more and more unpredictable. For instance, we would need to model the trajectory of the ant's joints to be able to predict the final pose. Bu,t by having options that control the displacement, given that we preserve information about where the ant starts we can predict where it will end. We leverage this to abstract details away from our ground state.
>
> - **Discrete vs. Continuous**:
> Although there are, indeed, discrete problems that are important, as previous work has tackled, not all problems we might have interest in solving are naturally discrete. For instance, the navigation problems in our experiments show that there is a continuity of $(x,y)$ positions to navigate. While discretization is definitely possible, it is not necessarily the most useful representation. Moreover, we also show that our approach is a strict generalization of the strong subgoal property that allows to build discrete models (as done for PDDL in previous work).
>
> - **Dataset** : yes! to build a good model we need a good coverage of the state-option space. In practice, what we do is to initialize the dataset with data from trajectories resulting from executing a random policy and, then continuously, enhance the dataset with trajectories obtained by the learning agent. This does not mean the agent uses these samples to learn, they are only used to update the model and the agent still plans in abstract imagination.

---

> ### Comment · Reviewer_NRqK · 2023-11-22
> **response to rebuttal**
>
> Thank you to the authors for responding. I believe there are some issues with unconvincing experiments brought up by a few reviewers, which are as of yet unresolved. I'll keep my score the same but encourage the authors to submit a new draft of the work to another venue. The required changes are significant enough that they warrant another round of reviews, imo.

---

### Official Review · Reviewer_36E8 · 2023-11-01

**Soundness:** 3 good
**Presentation:** 3 good
**Contribution:** 3 good
**Rating:** 8
**Confidence:** 4

**Summary:**

The paper presents an approach for learning dynamics preventing abstractions for sensorimotor observation space. Given a set of high-level skills and the learned dynamics preserving abstractions, the paper claims to develop an approach for planning for a solution.

The approach is evaluated in two test domains where the paper shows the visualization of the learned abstractions.

**Strengths:**

- For the most part of the paper, it is extremely well written. Given the wide use of embodied AI systems and robots, an approach that generates plannable abstractions for high-dimensional sensor input is extremely important.

- The paper nicely motivates the problem.

**Weaknesses:**

While the paper in general is nicely written, it has a few limitations:

- The paper advocates learning a continuous abstract  representation instead of a symbolic abstractions. However, it does not provide any reasons to that. Why are continuous abstractions more desirable than symbolic abstractions?

- Sec 4.1 is unclear. The notation for MI is a bit unclear. It needs to be made more clear. Sec 4.1 requires a re-writing including more explanation for the equation.

**Questions:**

I have two important questions:

- How is the dynamics preserving abstraction defined in Def. 3.6 different from the Markovian abstractions defined in [Srivastava et al. 2016]?

- Can you discuss the differences between the presented approach and [Allen et al. 2021]

Reference

Allen, Cameron, et al. "Learning markov state abstractions for deep reinforcement learning." Advances in Neural Information Processing Systems 34 (2021): 8229-8241.

Srivastava, Siddharth, Stuart Russell, and Alessandro Pinto. "Metaphysics of planning domain descriptions." Proceedings of the AAAI Conference on Artificial Intelligence. Vol. 30. No. 1. 2016.

---

> ### Author Response · Authors · 2023-11-16
>
> Thank you for your comments! Below the answers to your questions:
>
> - **Markov Abstractions**: Markov abstractions deal with finding compression functions that preserve Markovianity at the primitive action level in both transition dynamics and rewards. That is, as much compression as possible at the primitive action level and task level. When we introduce temporal abstractions, we lose information by only observing (s, o, s’) (as oppose to the entire trajectory generated by the option execution). Therefore, some details of the state become less and less predictable. We leverage that to abstract beyond the point of Markovianity (as defined in the original MDP) and we preserve *Markovinity at a higher time scale*. However, in that process we lose information about the ground state (given an abstract state we cannot know exactly in what ground state we would be in) and, hence, we cannot fully predict the ground reward function. And we don’t want to preserve the ground reward predictability! because we want to use our abstract model for new tasks that we can solve with our options.
>
> - How is the dynamics preserving abstraction defined in Def. 3.6 different from the Markovian abstractions defined in [Srivastava et al. 2016]?
> Thank you for pointing this reference out! I believe this shares the spirit of abstraction we are proposing in this paper: we can introduce impreciseness without being incorrect. Abstracting beyond the Markovianity of the ground MDP, introduces impreciseness on what ground state will be when an option is executed in the real environment, but preserves the relevant information to execute the next option. For instance, in the antmaze experiment, the model preserves knowledge of the ant’s position in the maze but cannot predict precisely with what pose the agent will reach such position.

---

> > ### Comment · Reviewer_36E8 · 2023-11-22
> > **Response**
> >
> > Thank you for the clarification.

---

### Official Review · Reviewer_dCJp · 2023-11-02

**Soundness:** 2 fair
**Presentation:** 3 good
**Contribution:** 2 fair
**Rating:** 3
**Confidence:** 4

**Summary:**

The paper introduces a method for enabling general-purpose agents to efficiently handle complex tasks by constructing abstract models based on temporally-extended actions. These models facilitate more efficient planning and learning and are characterized using principled conditions. The approach provides empirical evidence of improved sample efficiency in goal-based navigation tasks and offers theoretical support for information maximization strategies in abstract state representation learning.
The authors claim that they introduced a method for creating abstract world models that empower agents to plan effectively for goal-oriented tasks. The key idea is to allow agents to construct reusable abstract models for planning with specific skills. This is achieved by characterizing the state abstraction that ensures planning without any loss in simulation, meaning that planning with the learned abstract model can generate policies for the real world. The paper also provides theoretical support for the use of information maximization as a reliable strategy for learning abstract state representations.

**Strengths:**

- Good overview of the related work.
- Good description of motivations and intuitions.
- proper choice of environment settings.

**Weaknesses:**

Major:
- Some measures are used without definition,
- It seems that there exists a lot of inaccuracies and impreciseness in the theories and definitions. See all questions!

minor:
- typos:
last paragraph of the introduction "the *agents* needs", definition 3.5 "$s_{o}$" must be "$s_0$"
- writing:
Define the abbreviations before using them, e.g. "PDDL", "VAE"

There is a chance that I have not fully understood what this paper is trying to present.

**Questions:**

1- What is $P(s'|s,o)$ used in the paragraph right after definition 3.1?

2- An option $o$ is defined, and then you mention $T(s'|s,o)$ to define the transition probability of taking option $o$ in $s$? $T$ earlier was defined on action space $A$. How is it applied on options without showing the relationship of $I_o$ and $\beta_o$ with $s$ and $s'$ under option policy $\pi_o$?

3-the paper has defined "$\bar {\gamma} = \gamma ^{\tau (s,o)}$ is the abstract discount factor, $\tau: Z \times O \rightarrow [0,\infty)$, which consists of contradictory phrases. How is ${\tau (s,o)}$ but defined as a function of abstract variables $Z$ instead of $S$? Not clear what $\tau$ is. If based on definition 3.1, it is the option's execution time starting from $s$ taking option $o$, it is not clear how in definition 3.2 it becomes a map from $Z$ and $O$ to a non-negative real.

4- What does definition 3.4 mean? $ \Pi = {\pi \in \Pi : \pi(·|s) = \pi (·|z) \forall s \in z}$ says the probability of taking actions/options in $s$ should be equivalent to the probability of taking actions/options in abstract states. Transitions of taking actions in states might take you to another state $s'$ inside the similar abstract state $z$. How can the policies used for both abstract states and states be equivalent? Unless you are just discretizing the continuous state spaces based on the optimal policies that are already given. Lots of interchangeable usage of symbols here. Not precise and is hard to follow.

---

> ### Author Response · Authors · 2023-11-16
>
> Thank you for pointing all these issues out! We have updated the notation all across the paper and hopefully now it should be clearer. We believe this has helped improve a lot the framework's exposition!
>
> **1) and 2)** We use $T(s’|s,o) = Pr(s’|s,o)$. (we have updated this in the notation to use only T). Moreover, we use the expected-length model of options. Hence, what we’re saying is that we execute the option as a black box, and observe how long it took ($\tau$) and where it ended at s’, and we model the option’s mean duration and the observed transition probabilities $Pr(s’|s, o)$ independently. Hence, the effect of the initiation sets and termination conditions are subsumed by the model of options.
>
> **3)** We have corrected the notation. This was a typo! However, the option’s duration maps to a real because we are modeling the average duration when the option starts executing at s (over possible trajectories the option can take from $s\rightarrow s’$).
>
> **4)** We have removed this confusing definition because of impreciseness. However, we meant to say that we only consider policies that can be learned in the abstract model. Because in the abstract model we don't want to model $s$ fully then we need a restricted class of policies that only uses the information available at the abstract level.

---

> > ### Comment · Reviewer_dCJp · 2023-11-22
> >
> > Thank you for your feedback and for enhancing the manuscript. I appreciate the concept of utilizing mutual information maximization for acquiring abstract models. Nevertheless, I think there are substantial revisions yet to be addressed in this project. Consequently, I will maintain my current evaluation.

---

### Official Review · Reviewer_gKE9 · 2023-11-05

**Soundness:** 2 fair
**Presentation:** 3 good
**Contribution:** 3 good
**Rating:** 5
**Confidence:** 3

**Summary:**

This paper proposes a grounded abstract model formulation with a dynamic preserving abstraction. This abstract state representation (and model) guarantees not only accurate future predictions but also the bounded values in the abstracted rollouts. This paper then provides its implementation using contrastive learning to maximize mutual information between the future state, and the current abstract state and option. The results show that training DDQN in imagination using the abstract model improves the sample efficiency.

**Strengths:**

* The paper proposes a solid foundation of the abstract model that preserves dynamics and values.

* The paper is well written.

* The visualization in Figure 3 clearly shows that the abstract state representations focus on important features in the original observation space.

**Weaknesses:**

* The main focus of the paper is to show the efficiency of planning and learning when using the proposed abstract MDP. The experiments in the paper are a bit simple to showcase the benefits of the abstract model for planning. It would be stronger if the experiment was done in more complex environments with much longer-horizon tasks, such as AntMaze experiments (Hafner 2022) or robotic manipulation tasks [a].

* Similarly, the comparisons in Figure 5 are essentially between model-free RL (ground) and model-based RL (abstract), which does not seem fair. It might be fair to compare the proposed method with other model-based RL approaches, such as Dreamer and TD-MPC.

* Exhaustive comparisons to the alternatives to the dynamics preserving abstraction would be interesting, such as bisimulation.

* Some highly relevant works on temporally-extended models [a,b] are missing in the paper. Proper comparisons to these approaches are necessary.

[a] Shi et al. Skill-based Model-based Reinforcement Learning. CoRL 2022

[b] Zhang et al. Leveraging Jumpy Models for Planning and Fast Learning in Robotic Domains. 2023

**Questions:**

Please address the weaknesses mentioned above.


### Minor questions and suggestions

* Figure 1 may want to explain why abstract state representations and options are helpful for planning and learning. However, Figure 1 does not seem to help understand the paper. To understand this figure, we first need to know about options and abstract state representations, and how they simplify planning.

* In Section 4.2, it is unclear whether $\mathcal{L}^T_{\theta, \phi}$ is used to update $f_\phi$ or not.

* For multi-goal experiments in the paper, using the same amount of environment steps for the abstract planning and the ground baseline would make it easier to understand how better or worse a method is.

* The appendix could be included in the main paper for easier navigation.

* What is the difference between Figure 7 and 8?

* Training the abstract planning method longer in Figure 7 and 8 would be helpful to see how it learns. Using different x-scales for two methods is okay but it would be better to have the same scale.

* Many minor typos in the paper.


---

Thank you for author responses. I would love to see comparisons to Dreamer-like baselines, but couldn't find the results by the end of the rebuttal period. Thus, I keep my rating, borderline reject.

---

> ### Author Response · Authors · 2023-11-16
>
> - **Model-free vs. Model-based**:
> It is worth pointing out that the comparison between DDQN on the ground MDP and DDQN in the abstract MDP shows that “losing” the information from a Markov state does not impede the DDQN agent finding a optimal policy (the DDQN is only ever trained with trajectories from the abstract model). This is because the retained information is the only relevant information.
> We, however, understand the concern about the evaluation between model-free and model-based comparisons. *Currently, we are working on a Dreamer-like baseline that we will add to the draft asap*.
>
> - **Environment choice**: It is true that we can choose more intricate domains. However, I would argue that our Antmaze domain (even though not egocentric as the one referenced) shows something interesting: the state vector is an expert feature vector that is Markov by design. However, introducing skills for navigation allow us to build a model that abstract away (automatically) even more details and let the agent concentrate in the only important features for maze navigation (the ant's position in the maze).
>
> - **Dynamics-preserving vs. Bisimulation (Model-preserving abstractions)**:
> The main difference is that our dynamics-preserving abstraction does not impose conditions over the reward function.
> Our main desiderata is that we can re-use the abstract model to plan for different tasks and, therefore, the reward function of the ground MDP will be relevant to measure the cost (negative reward) of executing an option. This quantity, however, may be low-level trajectory-dependent and, hence, we might not be able to abstract our state beyond the Markov state of the ground MDP (e.g. we need to know the joints' states to compute the energy cost). Therefore, we decided to preserve only our ability to predict dynamics, and estimate the base reward of executing an option as the average reward accumulated (over the possible trajectories the option can generate).
> Hence, bisimulation can help reduce a high-dimensional observation (e.g. pixels) to a Markov state for the ground MDP. But to reduce our state even more, we use options and the dynamics-preserving condition.
>
> - **Missing references**: We have added this references to our related works sections! thanks!
>
> Minor questions:
>
> - Yes! $\mathcal{L}^T_{\theta, \phi}$ affects the learning of $f_\phi$ (corrected this sentence in the paper, thanks!)
> - There's not difference between figure 7 and 8 (removed).

---

### Author Response · Authors · 2023-11-16
**Paper draft update**

We thank the reviewers for their time and useful comments, that we believe have helped improve our paper.

We see across the reviews that there were concerns about the theoretical framework because of some notational issues and writing. We have updated the content of the paper with clearer and more consistent notation taking into account your comments and some additional suggestions we had received. Moreover, we have made explicit some parts of the framework’s construction that we had omitted and that should help make the theory clearer and easier to follow.

We have also updated our background section to include the expected-length model of options, upon which we build our framework, and we have rewritten the state abstraction background to make explicit the definition of state abstraction we build upon in this work: we use probabilistic groundings as our state abstractions, as opposed to state aggregation mappings as typically used.

Finally, we thank all the reviewers for pointing out relevant references that we have now included. Moreover, we have added an extra paragraph describing theoretical and foundational work in the state aggregation/abstraction in RL literature.

There’s also concern in regards the experimental section and the choices of baselines. Although it is true we haven’t compared with state of the art MBRL methods, we believe that our main contribution is in the novel approach to world model abstractions, which can be used in combination with more sophisticated planning methods such as TD-MPC and Dreamer-like agents and architectures. **However, we are currently working on new baseline with which we will update our draft as soon as we have them**

---

### Meta-Review · Area_Chair_Lr4p · 2023-12-05

**Metareview:**

**Summary**: The paper proposes a method for learning MDP state abstractions that preserve information needed for planning. This is achieved with an implementation based on contrastive learning to maximize mutual information between future states and the current abstracted state and option. Experiments showcase the method in two simple domains (pinball and antmaze). The paper also provides theoretical support for the use of information maximization as a reliable strategy for learning abstract state representations.

**Strengths**:
- The paper is well written.
- The paper addresses an important topic (abstraction learning) and is theoretically grounded.

**Weaknesses**:
- Unconvincing experiments: The experiments are done in rather simple environments with short-horizon tasks.
- Unconvincing comparisons: The baseline comparisons are limited, showcasing only a model-free RL vs model-based RL comparison. Reviewers proposed other baselines like Dreamer, TD-MPC, HIRO, HVF, DIrector but they were not ultimately addressed in the rebuttal.
- The paper is missing some related work pointed out by some of the reviewers.

Overall, most reviewers found the evaluation unconvincing due to simplistic environments and tasks, as well as due to unconvincing baseline choices. The authors promised new baseline comparisons during the rebuttal but never delivered them. As such, I think this paper is not yet ready for publication.

**Justification For Why Not Higher Score:**

Simplistic environments and not convincing enough baseline comparisons.

**Justification For Why Not Lower Score:**

N/A

---

### Decision · Program_Chairs · 2024-01-16

Reject